# Maternal Fiber Intake and Perinatal Depression and Anxiety

**DOI:** 10.3390/nu16152484

**Published:** 2024-07-31

**Authors:** Neda Ebrahimi, Tiffany Turner, Faith Gallant, Abinaa Chandrakumar, Roshni Kohli, Rebecca Lester, Victoria Forte, Kieran Cooley

**Affiliations:** 1Department of Research and Clinical Epidemiology, Canadian College of Naturopathic Medicine, Toronto, ON M2K 1E2, Canadakcooley@ccnm.edu (K.C.); 2Department of Research and Clinical Epidemiology, Canadian College of Naturopathic Medicine, Vancouver, BC V3L 5N8, Canada; 3NCCO Rehabilitation, Toronto, ON M4J 3S2, Canada; faithgallant.7@gmail.com; 4School of Public Health, University of Technology Sydney, Sydney, Ultimo 2007, Australia; 5National Centre for Naturopathic Medicine, Southern Cross University, Lismore 2480, Australia; 6Department of Human Biology, University of Toronto, Toronto, ON M5S 3J6, Canada

**Keywords:** dietary fiber, perinatal mental health, maternal diet, maternal depression, maternal anxiety

## Abstract

(1) Background: Dietary fiber can significantly alter gut microbiota composition. The role of the gut microbiome in the Gut–Brain Axis and modulation of neuropsychiatric disease is increasingly recognized. The role of antenatal diet, particularly fiber intake, in mitigating maternal mental health disorders remains unexplored. The objective of this review is to investigate the association between maternal fiber intake and perinatal depression and anxiety (PDA). (2) Methods: A literature review of PubMed and Google Scholar was conducted using appropriate keyword/MeSH terms for pregnancy, diet, fiber, and mental health. Observational and clinical trials published between 2015 and 2021 were included and data pertaining to dietary patterns (DP), food intake, mental health, and demographic data were extracted. The top three fiber-containing food groups (FG) per study were identified using a sum rank scoring system of fiber per 100 g and fiber per serving size. The consumption of these top three fiber FGs was then ranked for each dietary pattern/group. Mental health outcomes for each study were simplified into three categories of improved, no change, and worsened. The relationship between top three fiber FGs consumed within each DP and mental health outcomes was analyzed using Spearman’s correlation. (3) Results: Thirteen of fifty-two studies met the inclusion criteria. Ten (76.9%) studies assessed DPs (seven examined depression only, two examined depression and anxiety, and one examined anxiety only). Seven (53.9%) studies reported at least one significant positive relationship between mental health outcomes and DPs while three reported at least one negative outcome. Three (23.1%) studies compared intake of different food groups between depressed and non-depressed groups. In studies of DPs, the average consumption ranking of the top three fiber FGs bore a significant inverse association with mental health outcomes [r = −0.419 (95%CI: −0.672–−0.078)] *p* = 0.015. In studies comparing the intake of different FGs between depressed and non-depressed groups, the consumption of top-ranking fiber foods was higher in the non-depressed groups, but significantly higher in four of the ten high fiber FGs. (4) Conclusions: This study reframes findings from previously published studies of maternal diet and mental health outcomes to focus on fiber intake specifically, using a fiber ranking system. A significant correlation between lower intake of fiber and poorer mental health outcomes warrants further investigation in future studies.

## 1. Introduction

### 1.1. Perinatal Depression and Anxiety (PDA)

Postpartum depression and anxiety are common disabling health issues prevalent worldwide. The global prevalence of postpartum depression (PPD) is over 17% among postpartum women [1]. In Canada, postpartum anxiety (PPA) and PPD occur in 23% of new mothers [2]. A study conducted by Bowen et al. reported that 27% of pregnant Canadian mothers have major depression [3]. A 2018 Canadian Survey on Maternal Health found a prevalence rate of 17.9% and 13.8% in depression and anxiety symptoms within the 13 months postpartum, respectively [4]. A meta-analysis of 30 countries reported a 4.2% prevalence rate of clinically diagnosed depression and anxiety in the 24 weeks postpartum [5]. PPD has been shown to impair secure attachment patterns and imposes serious adverse effects on the physical and mental health of the mother–infant dyad [6,7]. Of specific note, a meta-analysis of 122 studies from the United Sated, Europe, Asia, Australia, and New Zealand found that PDD impacted the care a mother was able to provide for her children. It is anticipated that this could originate from early physical separation or a lack of material emotional availability. Authors noted that the quality of the mother–infant relationship had an impact on infant development, further affecting the relationship, causing a positive feedback loop [7]. In addition to impacting the mother–infant dyad, PDD has serious impacts on the well-being and quality of life of the mother. For instance, maternal mental health disorders continue to increase the risk of maternal suicide in Western countries including Canada [8]. Additionally, individuals with PPD have been found to be more likely to consult general practitioners, pediatricians, or mental health professionals for non-routine care, demonstrating the impact PDD can have on their overall health [7]. While treatment options for PPD include pharmacotherapy and psychotherapy, safety concerns and limited access to these treatment options pose barriers to implementation [9]. To address these barriers and consider patient preferences for treatments that align with their views on antidepressant use and breastfeeding [10,11], non-pharmacological approaches for PPD appear to be warranted.

### 1.2. Gut Microbiome and Mental Health

There is now a growing body of evidence supporting the connection between mental health and the gut microbiome, owing to the bidirectional communication pathways between the intestines and the brain (Gut–Brain Axis (GBA)). The GBA communication is mediated by several players including the immune system, neuroendocrine, hypothalamic–pituitary–adrenal (HPA) axis, short-chain fatty acids (SCFA), and autonomic, enteric, central nervous systems [12]. Gut health, decided by the microbial profile and their metabolic byproducts, known as the gut microbiome, is continuously modified by many environmental factors, such as diet, stress, smoking, drug use, etc. These factors can cause perturbance of the microbiome, leading to a more pathologic profile; this shift, referred to as microbial dysbiosis, is associated with a continuously expanding list of inflammatory and non-communicable diseases including several neuropsychiatric disorders [13]. Furthermore, microbial dysbiosis contributes to the permeability of the intestinal mucosa (leaky gut), causing an upregulated immune response resulting in chronic neuroinflammation over time [14]. Increased inflammatory biomarkers including inflammatory cytokines have indeed been found in patients with major depression and generalized anxiety disorder (GAD). Inflammatory cytokines can cross the blood–brain barrier and interact with pathophysiological processes involved in depression, including neurotransmitter metabolism, neuroendocrine function, and neural plasticity [15]. Recent studies have shown interesting trends in specific microbial species over others in patients suffering from major depressive disorder (MDD) and GAD [16]. In a retrospective study, Jiang et al. found a correlation between increased levels of family *Enterobacteriaceae* and *genus Alistipes*, but reduced levels of genus *Faecalibacterium* in major depressive disorder [17]. In a prospective study, Jiang et al. found that participants with GAD had significantly decreased microbial richness and diversity, a reduced number of bacteria that produce short-chain fatty acids, and overgrowth of the bacterial genera *Escherichia*, *Shigella*, *Fusobacterium*, and *Ruminococcus gnavus* species [18].

### 1.3. Diet, Microbiome and Mental Health

The relationship between diet and risk of depression and anxiety disorders has been investigated in numerous studies of non-pregnant adults. Healthy eating patterns containing fruits, vegetables, meats, fish, grains, and dairy products are shown to be associated with a lower likelihood of depression and anxiety [19,20].

Many features of a ‘healthy’ diet attributed to positive mental health outcomes may include the higher content of antioxidants, phytochemicals, vitamins, and minerals. Additionally, the greater consumption of whole fruits, grains, and greens, naturally exposes individuals to a greater amount of dietary fiber, which is a key modifier of the microbial profile. What remains unknown, however, is the direct impact of dietary fiber on the microbiome, the GBA, and mental health outcomes.

Dietary fiber is defined as plant-derived carbohydrates and includes non-starch polysaccharides, resistant oligosaccharides, lignin, and resistant starch [21]. Fiber has been recognized worldwide as an important staple of a healthy diet, yet most countries report inadequate fiber intake [22]. Fibers can be categorized as soluble and insoluble. Soluble fibers can dissolve in water and form a gel-like substance; they lower blood cholesterol and stabilize blood sugar levels. Insoluble fibers add bulk to the stool, aid in the prevention of constipation and maintenance of digestive health. Studies have revealed an interconnection between fiber and alteration of the gut microbiome and intestinal barrier. Microbiota actively metabolize fiber in the cecum and large intestine, where it remains unaltered by intestinal enzymes [21,23]. The byproducts of fiber fermentation by microbiota are short-chain fatty acids (SCFA), of which acetate, propionate, and butyrate are the most studied [21].

Various sources of fiber have been shown to change the strains of the gut bacteria [21]. A reduction in soluble fiber is linked to an alteration of microbial metabolites such as loss of phylum Bacteroidetes and an increase in class Clostridia and phylum Proteobacteria species [18].

*Dysbiosis* is marked by an increase in proinflammatory bacteria. A reduction in anti-inflammatory bacteria has been observed in MDD, particularly an increase in the phyla Bacteroidetes/Firmicutes ratio [15]. The increase in Bacteroidetes has been associated with depression-related intestinal inflammation [16]. In a clinical study, an increase in dietary fiber/prebiotics along with postbiotics like SCFA, increased the abundance of beneficial bacteria; while another study has shown Bifidobacterium strains possess anti-inflammatory effects by modulating tryptophan metabolism and 5-hydroxytryptamine (5-HT) synthesis [13]. These findings suggest that targeting intestinal microbiota as a measure to prevent and manage mental disorders should be further explored [17]. 

### 1.4. Microbiome in Pregnancy

The gestational period is associated with marked changes in the maternal gut and vaginal microbiome. The changes in microbiome composition occur throughout pregnancy and are most pronounced in the third trimester. Increases in genera *Akkermansia*, *Bifidobacterium*, and phylum Firmicutes are seen in parallel to the increased need for energy storage. Increases in proinflammatory bacterial phyla, such as Proteobacteria and Actinobacteria, are thought to have protective effects on the mother and the fetus [24]. In late pregnancy, there is an overall reduction in the gut microbiota, characterized by a decrease in the number of phyla Firmicutes and an increase in Proteobacteria, Actinobacteria, and genus *Streptococcus* [25]. Vertical transmission of bacteria from mother to infant is particularly important in establishing the infant gut microbiome and the development and maturation of their immune system. In the days after birth, the skin, mouth, and intestine of infants delivered vaginally will be populated by micro-organisms from the mother’s vaginal area, feces, breast milk, mouth, and skin. Initially after birth, the intestinal microbiota of the newborn is dominated by family *Enterobacteriaceae* and genus *Staphylococcus* but is later replaced by genus *Bifidobacterium* and some lactic acid bacteria [25]. The gut microbiota interacts with gut immune cells, establishing tolerance and dictating the development of inflammatory and autoimmune disorders. The first 1000 days of life is a critical period in the establishment of an infant/child’s microbiome and their subsequent long-term health outcomes. Thus, the health of the maternal microbiome during pregnancy and postpartum has a long-reaching impact, beyond maternal well-being [25]. As such, diet, particularly fiber, may play a crucial role to the health of the microbiome and consequently the health of mother and infant.

### 1.5. Gestational Diet, Microbiome, and PDA

Many studies have focused on maternal nutrition, pregnancy, and neonatal outcomes [26]. Few, however, have prioritized mental health outcomes. Commonly studied dietary patterns include the fertility diet, low carbohydrate diet, Western-type diet, Mediterranean diet, and Dietary Approaches to Stop Hypertension (DASH). A “health-conscious” dietary pattern, consisting of vegetables, fruits, nuts, pulses, fish and seafood, olive oil, and dairy products is protective against postpartum depressive symptoms. Another study exhibited that “traditional”, and “health-conscious” dietary patterns had a protective effect on anxiety symptoms. The relationship between maternal fiber intake and symptoms of depression and anxiety has not been investigated [22,23].

### 1.6. Study Objective

The objective of this review and evidence synthesis is to understand if a relationship between maternal fiber intake and mental health outcomes is present. Given the knowledge gap on the role of fiber specifically, in diet and PDA studies, we aim to re-examine published literature in the last 7 years to decipher the contribution of fiber, on mental health outcomes of pregnant women. This timeline was chosen so that the information collected was relevant to the most current changes/updates to dietary guidelines, fiber fortification of foods (snacks specifically), and general dietary trends subject to fads, information, and recommendations that affect dietary intake in different cohorts [27]. Our aim is to close the knowledge gap in the current literature and provide a new perspective on dietary studies in maternal mental health.

## 2. Materials and Methods

A literature review of PubMed and Google Scholar was conducted using keyword/MeSH terms: [diet, nutrition, dietary pattern, diet quality, fiber, prebiotic, oligosaccharides, complex carbs, prebiotics, symbiotic, fructooligosaccharides, inulin, oligofructose, galactooligosaccharide, xylooligosaccharides, vegetables, fruits, whole grains, legumes, fiber/fibre supplements, vegetarian] AND [mental health, anxiety, depression, mental illness, well-being, mood, stress, psychiatric disorders, psychological status, dysthymia, baby blues] AND [antenatal, pregnancy, postpartum, perinatal, peripartum, maternal, gestational age, lactation, breastfeeding].

Observational and clinical trials published since 2015 in pregnant and/or postpartum cohorts were included. Reviews, meta-analyses, studies prior to 2015, animal studies and studies of other mental health disorders were excluded. Article titles and abstracts were screened by three independent reviewers. Studies meeting inclusion criteria were reviewed, and variables related to diet, fiber intake, mental health outcomes, and demographic data were extracted.

The food items for each food group (FG) (i.e., grains, fruits, vegetables, nuts, etc.) in a study were extracted and evaluated for fiber content. Fiber content for every 100 g serving and typical serving size (TSS) of that food item was derived from food databases (i.e., United States Department of Agriculture (USDA), Canada Food and Nutrient Dataset (CFND), etc.). A fiber score (FS) was calculated by multiplying the fiber content per 100 g serving by the TSS; established cut points for % of recommended daily values were used as guides for FS (e.g., <5% of recommended daily value is considered ‘a little’) [28]. The FS and fiber per 100 g of all food item within a FG were averaged to calculate the FS and fiber/100 g for the given FG.

The FGs were then ranked according to highest FS and the highest fiber content per 100 g serving; the ranks were then summed to create a final fiber rank (FFR) per food group. Lower FFR corresponds to a higher fiber content. The three top-ranking FGs (i.e., FGs with the lowest FFR) for each dietary pattern in each study were identified and the relationship to mental health outcomes observed were reanalyzed using correlational statistics.

To our knowledge, comprehensive methods or resources to extrapolate fiber content for synthesis have not been established and up to 75% of dietary studies fail to capture fiber intake at all [29,30]. The next sections describe our process using hypothetical examples to estimate crude fiber exposure in each study.

In studies that did not make their food item lists available, the first and last and/or corresponding authors of studies were contacted to provide the list of foods, or the Food Frequency Questionnaires used in their studies to assess the diet in their cohorts. After three unsuccessful attempts, our reviewers used multiple government and industry websites to determine the most popular/typical food items within each food group. To identify what is most consumed from each food group in each country a combination of published literature, government websites, and a search of online popular supermarket brands were used. While the former sources were scarce and non-existent for most, identifying and searching popular brands proved to be exceedingly cumbersome and required sampling from several different popular stores and brands to come up with an estimated fiber and serving size for a given FG. Once this was compiled for each food group, we proceeded to calculate FFRs for each food group.

To account for the vast variation in serving sizes, we also ranked food groups by their fiber per 100 g serving. The sum rank of both (FS and fiber/100 g) were used to ultimately decide which 3 FGs had the highest fiber content. The following sections demonstrate this using hypothetical examples.

### 2.1. Determining Typical Serving Size (TSS)

As serving sizes vary by country, food and cuisine types, and personal preference, we defined typical serving size (TSS) as the medium size, volume, or quantity of any given food.

**Example** **1.**(Orange) *On the United States Department of Agriculture (USDA) website https://fdc.nal.usda.gov (accessed on 11 October 2023) the TSS for “Orange, all commercial varieties, raw”, is listed as 96 g, 131 g, and 184 g for one small, medium, and large orange, respectively, and 190 g for one cup of sectioned oranges. For our FS calculations we chose the medium size. Our assumption is that when eating an orange, the typical person peels and eats a medium size orange.*

**Example** **2.**(Pineapple) *For “Pineapple, raw, traditional varieties”, one pineapple weighs about 1 kg, 1 slice = 84 g, 1 cup diced = 174.4, and ½ cup diced = 85 g. To determine the FS, we used ¾ cup = 129 g as the ‘typical’ serving size. This is considered the in-between serving size between a full cup and ½ cup. The assumption is that when eating pineapples, the typical person will consume just under 1 cup of cubed pineapple.*

**Example** **3.**(Beans) *“Boiled, black, mature beans” are reported to weigh 91 g, 127 g, and 182 g for ½ cup, ¾ cup, and 1 cup on the Canadian Nutrient File database (CNFD) https://food-nutrition.canada.ca/cnf-fce/?lang=eng (accessed on 11 October 2023). For calculating FS, we used the 127 g = 175 mL or ¾ cup as the TSS of edible beans.*

**Example** **4.**(Prepared Dishes) *In bean-based dishes or rice-based dishes (i.e., curry, seafood fried rice, chili, etc.), we examined common recipes listed on the USDA, CNFD, and/or popular restaurant websites that had published nutritional information by serving size. For example, the most common serving size reported for chili is 1 cup. Thus, we used 1 cup as the TSS which is reported to weigh 236–267 g on USDA website for different chili dishes https://fdc.nal.usda.gov/fdc-app.html#/ (accessed on 11 October 2023). We used the average of this range to determine FS for chili (and other prepared dishes in a similar manner).*

### 2.2. Calculating Fiber Scores (FS)

For any given FG, the average fiber per 100 g serving (edible portion, no refuse) and the TSS of all food items in that FG, were inputted. For example, if a study included pineapples, bananas, and oranges in their fruits FG, the fiber/TSS is calculated by multiplying the fiber/100 g serving by the TSS for that fruit.

Fiber scores (FS) are then assigned based on fiber/TSS values. The FS is a nominal value between 1 and 4. Less than 1 g of fiber/SS is a score of 1 and is defined as ‘very low’, 1 to <2 g corresponds to a FS of 2 and is defined as ‘low’, 2 to <5 g correspond to a score of 3 and is defined as ‘moderate’, and anything containing 5 g or more corresponds to a score of 4 and is defined as ‘very high’. In the absence of standard definitions for high, medium, and low fiber, the FS definition and values were arbitrarily chosen by the authors, but consistent with typical dietary definitions (See Table 1).

Table 2 demonstrates how FS for the ‘Fruits Food Group’ in a hypothetical study would be calculated using fruit items: pineapples, bananas, and oranges. The FS for the FG fruits in this example is determined by averaging the fiber/TSS of all included fruits. In this example, an average fiber/TSS of 2.36 corresponds to a FS = 3 (Table 1) and is defined as moderate level of fiber content.

### 2.3. Calculating Final Fiber Ranks (FFR)

After calculating FS for each of the FGs in a study, we then ranked the FGs according to fiber content per 100 g as well as fiber score. The sum of both ranks was used to create a final fiber rank (FFR) for the given FGs. The top-ranking fiber FGs (i.e., lowest FFR) were then used for our analysis. Table 3 demonstrates the ranking process. In Table 3, the highest fiber ranking FGs are: (1) legumes, (2) nuts, and (3) fruits and (4) cereals/grains. The fruits and cereals/grains tied in 3rd place. After ranking FGs according to FFR and identifying the highest fiber ranking FGs (i.e., FGs with the lowest FFR), we analyzed their consumption within each dietary pattern in the study.

### 2.4. Consumption Ranking of Highest Fiber FGs in Each DP

Table 4 shows a hypothetical study, in which 7 FGs are sorted from most to least consumed in each of the three identified dietary patterns. The consumption ranking of the highest fiber containing FGs, legumes (FFR = 1) and nuts (FFR = 2), are analyzed within each DP. In this example, legume consumption is ranked 5th, 1st, and 7th in DP-1, DP-2, and DP-3, respectively; whereas nut consumption is ranked as 7th, 3rd, and 5th in the same DPs.

Given that different number of FGs are analyzed in each study, we express the consumption ranking as percentage ranks. In this example 7 FG are included, so the percent consumption rank for legumes is 71.4%, 14.2%, and 100% in DPs-1, DP-2, and DP-3, respectively; and for nuts, 100%, 42.9%, and 71.4% in the same order DPs. Hence, only DP-2 has the highest consumption for the highest fiber FGs in this example.

### 2.5. Simplifying Mental Health Outcomes

We then examined the association between reported mental health outcomes in relation to the consumption ranking and percent consumption ranking of the highest fiber FGs within the DPs.

To do this, we simplified reported statistically significant outcomes for anxiety and depression in each study as ‘same’ (Score = 0), ‘improved’ (Score = +1), and ‘worse’ (Score = −1) for each DP. For studies where asynchronous findings were reported for anxiety and depression, the net score was used to represent the overall mental health score. For example, if one outcome worsened and the other improved, the net effect is treated as zero (i.e., no change), and if one worsened while the other did not change, the net effect would be scored a ‘−1’ (worse), and if one improved and the other did not change, the net score for mental health would be ‘+1’ (improved).

### 2.6. Statistical Analysis

Given the ordinal nature of the outcome variable (mental health Score: 1, 0, −1), we used Spearman’s correlation to evaluate the relationship between consumption rank and percent consumption rank of the top 3 fiber FGs and the simplified mental health outcomes within each DP. The 95% confidence intervals, Spearman’s rho, and two-tailed significances were reported for each relationship.

## 3. Results

A total of 53 studies were identified in the initial database searches. After duplication removal, fifty-one studies were screened of which twenty-eight were omitted (fifteen studies were published prior to 2015, four reviews, four missing maternal mental health outcomes, four focused on dietary quality/behavior/diversity, and one non-pregnant cohort). A total of twenty-three studies appeared eligible for full text review, of which ten were later excluded due to significant challenges in identifying food groups/items (six), wrong direction of association (one), comparing specific food groups only (one), and extreme poverty/food insecurity (two) at the country of study (Figure 1). Thirteen studies were included for the final analysis [31,32,33,34,35,36,37,38,39,40,41,42,43]. 

Appendix A summarizes the characteristics of the studies analyzed. Ten (76.9%) studies analyzed mental health outcomes in relation to DPs, and three (23.1%) studies compared intake of different FGs between depressed and non-depressed cohorts.

Appendix A lists the top three fiber ranking FGs in each study and their consumption ranking within each dietary pattern. The total number of FGs/items included in the study, and the relative placement of the top three fiber FGs (percentile placement) are also demonstrated. Additionally, the significant mental health findings from Appendix A, is simplified to worsened, unchanged, and improved categories for each DP.

For example, in study #3 [40] of the 33 FGs included, the highest-ranking fiber FGs are seaweeds, mushrooms, and beans. Three dietary patterns, healthy, Japanese, and Western were identified in this study. In the healthy DP, seaweeds ranked fifth (5/33 = 15.2%), mushrooms third (9.1%), and beans ranked fourth (12.1%), suggesting that in the healthy DP, the top three fiber ranking FGs were commonly consumed. By comparison, in the Western DP, the same three FGs were the least consumed FGs.

Statistical tests assessing the relationship between the consumption ranking, consumption ranking percentiles, and overall mental health changes within each DP, were analyzed using Spearman’s correlation. Table 5 illustrates this analysis. A strong inverse correlation was found between the consumption ranking of the first [r = −0.41 (95%CI: −0.66 to −0.06) *p*-value: 0.019] and third [−0.46 (95%CI: −0.696 to −0.122) *p*-value = 0.008] and average ranking of all top three [rho = −0.419 (95%CI (−0.67 to −0.078), *p*-value: 0.015] fiber FGs in relation to mental health outcomes. The same finding was observed for the percentile ranking and average top three percentiles (See Table 6). The second highest fiber FGs did not bear any significant relation to mental health outcomes.

## 4. Discussion

The correlation between the microbiome and neuropsychiatric disease has been demonstrated in the literature and is largely accepted in the scientific community. Numerous studies have linked poor diets to poor mental health outcomes, and since the gut microbiome is greatly impacted by diet composition, it is of interest to understand the role it may have in modifying mental health outcomes.

Plant fibers are indigestible carbohydrates that can only be metabolized by specific species of gut microbiota via anaerobic fermentation, the primary product of which is SCFA. It is well established that different fibers can alter the microbiome profile (and output) and exert effects on the host. The type of effect depends on the physiochemical properties of the ingested fiber [44].

The therapeutic potential of fiber in mental health, however, has received little attention. No studies at the onset of this review had investigated the relationship between maternal fiber intake, gut microbiome, and perinatal mental health outcomes. The few relevant studies on this topic are limited to maternal nutritional status, macronutrient intake, dietary patterns, and dietary quality and the subsequent impact on, primarily, depression.

This study is the first to focus on fiber intake and perinatal maternal anxiety and depression. The major challenge for this review was the absence of fiber data, and the need to use proxy variables to assess fiber exposure in each study. We used a ranking system in each study, by which we identified the highest fiber FGs, and ranked their consumption within each of the dietary patterns in that study. We then simplified the mental health outcomes in each study and assessed this in relation to the consumption ranking of the top three fiber FGs within each dietary pattern. In doing so, we reframed the findings for the dietary patterns/intakes to fiber intake. Analyzing this relationship yielded the results that higher consumption of the highest fiber FGs was negatively correlated with mental health outcomes.

Without a list of food items for the studied FGs, standardized serving sizes, and the intercultural/continental variations in both, many arbitrary assumptions needed to be made. This may be one reason why the ranks for fiber FGs vary amongst the studies. A country with a higher consumption of white rice will have a lower fiber ranking in their grains/cereals FG than a country with a higher consumption of whole grain breakfast cereals. Likewise, a country with a heavy consumption of beans in their traditional dishes will have a higher fiber ranking for their prepared-dish FG than one with a higher noodle consumption.

To account for the vast variation in serving sizes, we also ranked food groups by their fiber per 100 g serving. The sum rank of both (FS and fiber/100 g) were used to ultimately decide which three FGs had the highest fiber content (Table 6). Given the scarcity and cumbersome nature of searching published literature, government, and retail websites just to identify popular brands and FGs in each country, we are confident that our approach is unique and helps consolidate gaps in the literature, regarding the consumption of dietary fiber in different DPs, and the correlation to mental health outcomes in mothers.

These assumptions and estimates of the most frequently consumed food items as well as serving sizes are the primary limitation of this study. Other limitations include the timing, frequency, and tools used to capture dietary intake and mental health outcomes in each cohort. Our approach to categorize mental health outcomes facilitated our ability to synthesize the existing literature. However, it prevents a more precise examination of magnitude or clinically meaningful associations between mental health outcomes and interventions with fiber-based dietary components.

The timing of assessments may be critical in the outcomes observed. For example, the risk of depression and anxiety may be higher in the early weeks postpartum than six months postpartum, and studies that did not assess outcomes in the first three months postpartum may have missed those early episodes [45,46].

Having a history of mental health disorders is a significant predictor of perinatal anxiety and/or depression, yet most studies did not assess or report this history in their cohort [46]. The use of antidepressants and psychotherapy, which can modulate disease courses, was also not consistently assessed or reported.

In the studies involving dietary patterns, most often the highest quartile was compared to the lowest quartile, yet some studies used one identified DP as the reference DP (i.e., Study 6 [33] and Study 24 [38]) to which others were compared to. This type of comparison may introduce confounders given the overlap between dietary patterns, and the lack of evidence for the sub/superiority of the reference DP.

Finally, the role of diet in mental health is increasingly seen as a synergistic play between macronutrients, minerals, vitamins and antioxidants, and foods typically higher in fiber tend to be more nutrient dense. Thus, the reported inverse association in this study, between the consumption of high fiber foods and mental health, is not of great novelty or may be confounded by other aspects of nutrition in food consumption. However, a focus on fiber intake specifically and mental health outcomes, may be warranted, as the primary modulator of gut microbiome, and the irrefutable link to anxiety and depression. A large observational study published in April 2022 identified fiber, some vitamin Bs, and magnesium as the primary drivers of mental well-being during pregnancy [47]. The mechanism by which the microbiome is involved requires further investigation.

Future studies should aim to quantify fiber intake during pregnancy and postpartum from all sources, including snacks, replacement meals (nutritional bars, supplements), and prebiotic supplements, using repeated assessments throughout the perinatal period. It will be of great interest to use a clinical population at risk of perinatal anxiety and depression, and to collect stool and blood samples in parallel to dietary assessments, to understand the impact on the microbial profile and output. Finally, mental health assessments should be conducted at least once every 3 months from early pregnancy until 12 months postpartum to ensure the capturing of all critical phases of the perinatal period, i.e., nausea and vomiting in the early trimester, weight gain, physical discomfort and sleep issues in later trimesters, delivery, breastfeeding, and recovery in the first month postpartum, etc.

Fiber intake is low in pregnancy across most pregnant populations. In Canada, prenatal fiber intake in one large cohort (N = 861) was a median 23.5 g/day, ~17% below the recommended 28 g/day [48]. If the therapeutic potential of fiber and prebiotic foods and supplements in mental health is established, diet alone can provide an accessible, effective, safe and affordable option to women everywhere, particularly those at risk of experiencing PDA. Future research, including clinical trials, is warranted.

## Figures and Tables

**Figure 1 nutrients-16-02484-f001:**
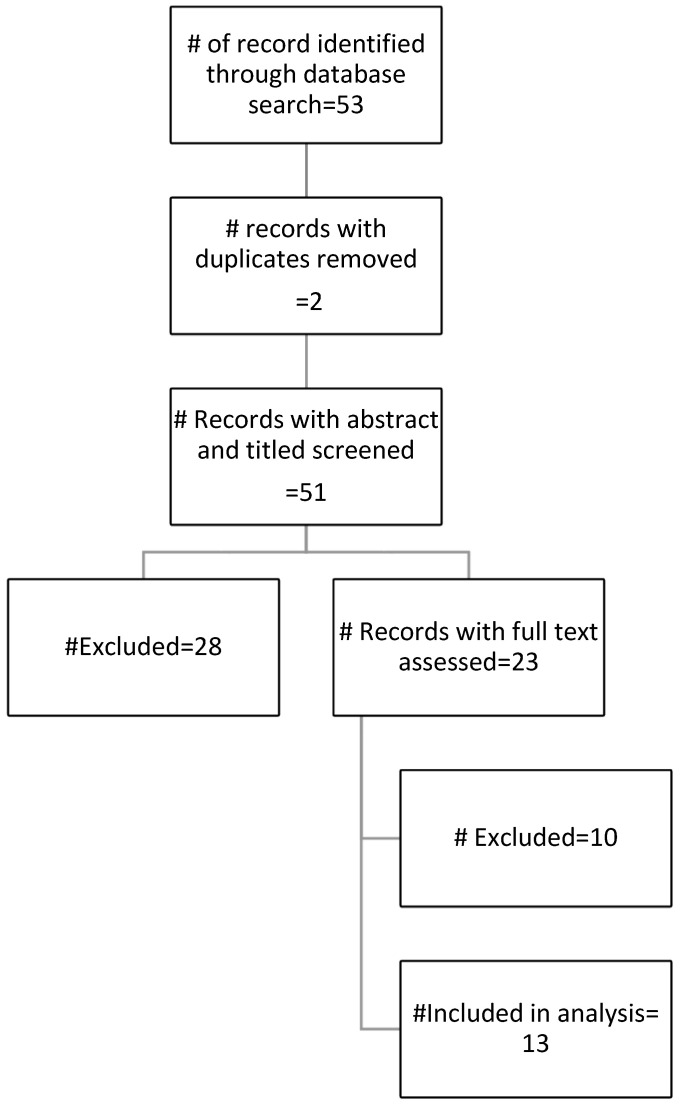
Literature search results.

**Table 1 nutrients-16-02484-t001:** Defining fiber scores.

Fiber/Serving Size (grams)	Definition	Fiber Score
<1 g	Very Low	1
1 to <2 g	Low	2
2 to <5 g	Moderate	3
5+ g	Very High	4

**Table 2 nutrients-16-02484-t002:** Example of FS calculation for the food group fruits.

Food Group Items	Fiber/100 g	TSS (g)	Fiber/TSS	Fiber Score
Pineapple	1.4 g	129 g	1.8 g	2
Banana	1.7 g	118 g	2.01 g	3
Orange	2.5 g	131 g	3.27 g	3
Fruits Food Group			2.36 g	3

**Table 3 nutrients-16-02484-t003:** Deriving final fiber ranks (FFR) for each food group.

FGs	Fiber/100 g	Fiber/100 g Rank	FS	FS Rank	Sum of Ranks	FFR
Fruits	1.87	4	3	2	6	3
Legumes	3.5	1	4	1	2	1
Nuts	3	2	2	3	5	2
Cereals and Grains	2.7	3	2	3	6	3

**Table 4 nutrients-16-02484-t004:** Consumption of highest fiber ranking FGs in each dietary pattern.

Consumption Ranking	% Consumption Ranking	Dietary Pattern-1	Dietary Pattern-2	Dietary Pattern 3
1	14.2	Seafood	Legumes *	Soda
2	28.6	Fruits	Fruits	Cereals and Grains
3	42.9	Meats and Poultry	Nuts *	Fruits
4	57.1	Sodas	Seafood	Seafood
5	71.4	Legumes *	Cereals and Grains	Nuts *
6	85.7	Cereals and Grains	Meats and Poultry	Meats and Poultry
7	100	Nuts *	Sodas	Legumes *

* Highest fiber containing FGs. (Legumes FFR = 1; nuts FFR = 2).

**Table 5 nutrients-16-02484-t005:** Relationship between consumption ranking and consumption ranking percentage of high fiber food groups and mental health outcomes.

Confidence Intervals of Spearman’s Rho				
Top 3 FG	Spearman’s Rho	95% Confidence Intervals (2-Tailed) ^a,b^		*p*-Value ^c^
		Lower	Upper	
1st Ranked	−0.407	−0.664	−0.064	0.019
2nd Ranked	−0.063	−0.407	0.296	0.727
3rd Ranked	−0.455	−0.696	−0.122	0.008
Average of 1st, 2nd, and 3rd	−0.419	−0.672	−0.078	0.015
1st % Ranking	−0.501	−0.726	−0.181	0.003
2nd % Ranking	−0.095	−0.433	0.267	0.599
3rd % Ranking	−0.454	−0.695	−0.120	0.008
Average of 1st, 2nd, and 3rd %s	−0.556	−0.760	−0.253	0.001

a. Estimation is based on Fisher’s r-to-z transformation. b. Estimation of standard error is based on the formula proposed by Fieller, Hartley, and Pearson. c: Two-tailed significance. Dependent variable: overall mental health outcome, independent variables: ranking and ranking percentile of highest fiber FGs within each DP.

**Table 6 nutrients-16-02484-t006:** Intake of top 3 fiber containing foods/food groups, between depressed and non-depressed patients.

			Depressed	Non-Depressed	Antidepressant Treated
Study #	# of Food Groups/Items	Top 3 Fiber FGs	Top 3 Rank (z-Score)	Top 3 Rank (z-Score)	Top 3 Rank (z-Score)
Galbally, 2021 [36]	9	Cereals	5 (−0.80)	6 (−0.85)	6 (−0.79)
		Fruit	3 (0.51)	3 (0.79)	3 (0.66)
		Bread	4 (0.00)	4 (−0.05)	4 (−0.05)
Avalos, 2020 [43]	12	Whole Grains	4 (0.44)	4 (0.63)	
		Fatty Acids	5 (0.05)	5 (0.17)	
		Greens and Beans **	12 (−1.30)	11 (−1.15)	
		Total Fruits **	10 (−0.96)	9 (−0.83)	
Shi, 2020 [35]	14	Staple Foods-Wheat	11 (−1.25)	11 (−1.21)	
		Other Vegetables **	1 (3.28)	1 (3.89)	
		Light Vegetables **	4 (0.98)	4 (1.34)	

Z-scores calculated using the means and standard deviations in the depressed groups; ** indicate significant difference in consumption between cohorts.

## Data Availability

The raw data supporting the conclusions of this article will be made available by the authors on request.

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
