# Peer review of "Maternal Fiber Intake and Perinatal Depression and Anxiety"

_nutrients, 2024, doi:10.3390/nu16152484_

Round 1

Reviewer 1 Report

Comments and Suggestions for Authors

The authors study the association between maternal fiber intake and perinatal depression and anxiety using a metaanalysis approach. They examined 13 studies and found a significant inverse association between high fiber food groups and maternal mental health. This is an important topic and an interesting approach.

As the 13 studies come from different parts of the world, the dietary patterns are very heterogeneous. The authors have taken efforts to rank the food groups by dietary fiber intake in a unique approach. The manuscript is well written and comprehensive in the introduction, methods and discussion. 

One aspect which was not discussed is in table 7, the second highest ranked and percent ranked food group did not show significant relationship with mental health outcomes. Would the authors have any postulation for this discrepancy?

Author Response

Comment 1: One aspect which was not discussed is in table 7, the second highest ranked and percent ranked food group did not show significant relationship with mental health outcomes. Would the authors have any postulation for this discrepancy  

Response 1: It is interesting that only that the 1st and 3rd highest ranked fiber food groups bore a significant relationship. We think this may be a shortfall of strategy employed in identifying most consumed foods and the assumptions made in serving sizes in the absence of both data. While our strategy provides the best educated guess, it is possible, even probable, that they are not accurate, and leading to the observed discrepancy. Inclusion of specific food items, along with their serving sizes in each study, would allow for a more accurate categorization and analysis of highest fiber containing food groups and mental health scores.

Reviewer 2 Report

Comments and Suggestions for Authors

Comment on “Maternal Fiber Intake and Perinatal Depression & Anxiety”

This is an interesting review exploring the relationship between maternal diet and postpartum mental health disorders. The work is well written and clear, Materials and methods are explained in detail and tables are well made. The paper can be accepted with the following few corrections: 

Paragraph “1.2. Gut Microbiome & Mental Health”

In this paragraph please cite and discuss the following articles in order to provide the reader a brief overview of the relationship between immune system, gut microbiome and obstetric outcome. 

·      Masucci L, D'Ippolito S, De Maio F, Quaranta G, Mazzarella R, Bianco DM, Castellani R, Inversetti A, Sanguinetti M, Gasbarrini A, Scambia G, Di Simone N. Celiac Disease Predisposition and Genital Tract Microbiota in Women Affected by Recurrent Pregnancy Loss. Nutrients. 2023 Jan 1;15(1):221. doi: 10.3390/nu15010221. PMID: 36615877; PMCID: PMC9823693.

In the discussion please cite and discuss the following articles explaining the importance of the diet in fertility:

·      Cristodoro M, Zambella E, Fietta I, Inversetti A, Di Simone N. Dietary Patterns and Fertility. Biology (Basel). 2024 Feb 19;13(2):131. doi: 10.3390/biology13020131. PMID: 38392349; PMCID: PMC10886842.

Finally, I suggest, if possible, to add an infographic image summarizing the different points brilliantly discussed by the authors in the different paragraphs.

Author Response

This is an interesting review exploring the relationship between maternal diet and postpartum mental health disorders. The work is well written and clear, Materials and methods are explained in detail and tables are well made. The paper can be accepted with the following few corrections: 

Paragraph “1.2. Gut Microbiome & Mental Health”

Comment 1. In this paragraph please cite and discuss the following articles in order to provide the reader a brief overview of the relationship between immune system, gut microbiome and obstetric outcome.

 Masucci L, D'Ippolito S, De Maio F, Quaranta G, Mazzarella R, Bianco DM, Castellani R, Inversetti A, Sanguinetti M, Gasbarrini A, Scambia G, Di Simone N. Celiac Disease Predisposition and Genital Tract Microbiota in Women Affected by Recurrent Pregnancy Loss. Nutrients. 2023 Jan 1;15(1):221. doi: 10.3390/nu15010221. PMID: 36615877; PMCID: PMC9823693.

Response 1. As this study is specifically focused on maternal anxiety and depression, we have made every effort to highlight the relationship between diet/microbiome and mental health, while adhering to the word count. The reference provided focuses on pregnancy loss/outcomes, and Genital Microbiome, which we don’t believe is relevant to the scope of our study.

Comment 2. In the discussion please cite and discuss the following articles explaining the importance of the diet in fertility: Cristodoro M, Zambella E, Fietta I, Inversetti A, Di Simone N. Dietary Patterns and Fertility. Biology (Basel). 2024 Feb 19;13(2):131. doi: 10.3390/biology13020131. PMID: 38392349; PMCID: PMC10886842.

Response 2. In line with the previous response, we have included references that support the theme of our study, which is fiber intake in pregnancy and perinatal anxiety and depression. The impact on fertility while an important topic is not relevant to the topic discussed.

Comment 3. Finally, I suggest, if possible, to add an infographic image summarizing the different points brilliantly discussed by the authors in the different paragraphs.

Response 3. The authors have taken great effort to describe the very complex and creative way of  extracting fiber information from each study, in the absence of data. Creating an infograph for  a review of this nature is highly complex and requires more sophisticated platforms. We will take the reviewer's suggestions into consideration and attempt to make one, however this will not be submitted, until the 2nd round of reviews. 

Reviewer 3 Report

Comments and Suggestions for Authors

Here are my suggestions and comments to improve your work:

1. Line 41, please explain the acronym PPD.

2. 1.1 paragraph, please expand with more reference and detail the paragraph.

This pragraph is not exaustive.

3. In lines 60-62 please insert even the Enteric Nervous System, this system has a great role in GBA communications.

4. Tables 5 and 6 please detail every study and highlight the differences between the other studies.

5. Line 326, Why the authors did not insert tables 5 and 6 in the paper?

If the Journal has a limit of tables, you should consider putting a literature search and the methodological table with examples like tables 1; 2 and 3 in the supplementary.

Comments on the Quality of English Language

none

Author Response

1. Line 41, please explain the acronym PPD.  Thank you for your comment. We agree and this comment has been defined on line 41.  2. 1.1 paragraph, please expand with more reference and detail the paragraph.This paragraph is not exhaustive. Thank you for your comment. We agree and have elaborated on this paragraph. Both further commentary on originally included references and additional references have been added. Changes can be found on lines 49 - 63.  3. In lines 60-62 please insert even the Enteric Nervous System, this system has a great role in GBA communications. Thank you for your comment. We agree and this change can be found on lines 72-73.   4. Tables 5 and 6 please detail every study and highlight the differences between the other studies. Thank you for this suggestion. We believe that all of the pertinent details of the studies that relate to our objective are contained in Tables 5 and 6
5. Line 326, Why the authors did not insert tables 5 and 6 in the paper? Thank you for this comment. Due to the cumbersome nature of these tables, they could not be embedded in a way that adhered to formatting and allowed the reader to easily read. They have been submitted as supplemental materials. 

Round 2

Reviewer 3 Report

Comments and Suggestions for Authors

The authors did not response exaustevely to my questions ( point 4 and 5), I recommend to insert the table mentioned in the text and insert detailed description of the studies included.

Author Response

Thank you for your comments. 

As per the MDPI style guide: “Very large tables, or many different tables showing similar cases, may be included in an Appendix or as supplementary data” https://www.mdpi.com/authors/layout. The non-specific request for what type of details are missing has made it challenging for authors to respond. Currently, the table lists size of participant population, country of origin, and descriptions of outcomes, and statistically significant results. What has been added to the tables is the first author last name and year of publication to allows readers to more easily identify the article. All included articles are observational studies. The authors wish to restate that they believe these details are appropriate and consistent with our study design and objectives.